# PLGA-Gold Nanocomposite: Preparation and Biomedical Applications

**DOI:** 10.3390/pharmaceutics14030660

**Published:** 2022-03-17

**Authors:** Alaaldin M. Alkilany, Ousama Rachid, Mahmoud Y. Alkawareek, Nashiru Billa, Anis Daou, Catherine J. Murphy

**Affiliations:** 1College of Pharmacy, QU Health, Qatar University, Doha 2713, Qatar; orachid@qu.edu.qa (O.R.); nbilla@qu.edu.qa (N.B.); adaou@qu.edu.qa (A.D.); 2Biomedical and Pharmaceutical Research Unit, QU Health, Qatar University, Doha 2713, Qatar; 3Department of Pharmaceutics and Pharmaceutical Technology, School of Pharmacy, The University of Jordan, Amman 11942, Jordan; m.alkawareek@ju.edu.jo; 4Department of Chemistry, University of Illinois at Urbana-Champaign, 600 South Mathews Avenue, Urbana, IL 61801, USA

**Keywords:** gold nanoparticles, poly(lactic-co-glycolic acid), PLGA, composite

## Abstract

A composite system consisting of both organic and inorganic nanoparticles is an approach to prepare a new material exhibiting “the best of both worlds”. In this review, we highlight the recent advances in the preparation and applications of poly(lactic-co-glycolic acid)-gold nanoparticles (PLGA-GNP). With its current clinically use, PLGA-based nanocarriers have promising pharmaceutical applications and can “extract and utilize” the fascinating optical and photothermal properties of encapsulated GNP. The resulting “golden polymeric nanocarrier” can be tracked, analyzed, and visualized using the encapsulated gold nanoprobes which facilitate a better understanding of the hosting nanocarrier’s pharmacokinetics and biological fate. In addition, the “golden polymeric nanocarrier” can reveal superior nanotherapeutics that combine both the photothermal effect of the encapsulated gold nanoparticles and co-loaded chemotherapeutics. To help stimulate more research on the development of nanomaterials with hybrid and exceptional properties, functionalities, and applications, this review provides recent examples with a focus on the available chemistries and the rationale behind encapsulating GNP into PLGA nanocarriers that has the potential to be translated into innovative, clinically applicable nanomedicine.

## 1. Introduction

Currently, the clinical practice hosts an increasing number of FDA-approved nanotechnology-based therapeutics and more are currently being investigated in clinical trials [1]. The latest example that demonstrates the importance of nanomedicine is the development of a range of successful COVID-19 vaccines during the current pandemic. Without the appropriate nanocarrier system, mRNA delivery to the cells is challenging. Moreover, nanoparticle-based therapeutics and imaging proxies are increasingly attracting attention from both academic and pharmaceutical perspectives, with an increasing number of publications and nanoparticle-based products in the market.

For the proper design of effective and safe nanotherapeutics, it is essential to understand how nanoparticles interact with biological compartments, referred to as the nano-bio interface, throughout this review. The nano-bio interface encompasses biological processes such as the formation of protein corona on circulating nanoparticles, cellular uptake/efflux, intracellular trafficking, and the pharmacokinetics (absorption, distribution, metabolism and excretion) of nanotherapeutics. Understanding these processes at the molecular level is vital and allows the engineering of optimal nanotherapeutics with maximum efficacy and minimal side effects, henceforth, resulting in higher rates of translation into the clinic.

To understand the nano-bio interface, nanoparticles should be visualized and quantified in complex biological compartments to answer fundamental questions such as “what pathways do nanoparticles follow and at what quantity are they delivered?”, “what is the effect of nanoparticles or their cellular-related parameters on their biological fate?” and many other questions one may have. Answers to these questions require the utilization of a wide range of analysis. These include the tracking, visualization and quantification of nanoparticles using available analytical platforms with a high level of sensitivity, selectivity and precision. Accordingly, a wide range of nanoparticle labeling techniques using both in vitro and in vivo models have been developed, including labels or probes that support fluorescence, magnetic resonance, computational tomography, positron emitting tomography, surface enhanced Raman spectroscopy, radionuclide, photoacoustic and electron microscopy imaging. The first generation of these labels, which are currently the most common, are small organic molecular labels such as chromophores, fluorophores and radionucleotides. Recently, inorganic nanoparticles have been employed as contrasting agents or probes in order to enhance the contrast capability and address limitations of small molecular labels including photobleaching, chemical degradation and desorption-related background signals. One example is gold nanoparticles (GNP) in immunogold (antibodies labeled with GNP). This is a typical biochemical assay that enables the visualization and confirmation of the presence of cellular and tissue antigens at great resolution (the limit of electron microscope resolution) [2]. Quantum dots labeled antibodies are currently available through commercial suppliers with various advantages over small fluorophore-labeled antibodies, including superior photostability and a lower tendency of leaching from the host antibodies [3].

With the many existing types of “nanoprobes” [4,5,6] and the material they are formulated from, and for the purpose of covering the topic thoroughly in a reasonable length, this review focuses on GNP as probes to label polymeric drug carriers (Figure 1). The fascinating advantages of GNP as an excellent probe coupled with the importance of understanding the nano-bio interface of polymeric drug carriers and their clinical relevance motivated us to write this review, which will:(1)Discuss the outstanding properties of gold nanoprobes to justify their use;(2)Discuss current available methods for gold nanoprobe’s surface functionalization and subsequent encapsulation into polymeric carriers;(3)Provide examples from the research work of our own and other groups on how labeling with gold nanoprobes allows for the precise tracking, visualization and quantification of pharmaceutical polymeric nanocarriers as well as the fabrication of laser-responsive drug delivery systems as summarized in Figure 1.

## 2. Brilliant Optical Extinction of Encapsulated GNP Enables Outstanding Sensing and Tracking Capabilities

At the nanoscale level, gold exhibits different physical, chemical, optical and biological properties compared to their bulk due to different photon-electron interactions at the nanoscale level when compared to its bulk counterpart [7,8]. While bulk gold reflects light and appears a bright gold, GNP absorb and scatter light and thus a suspension of GNP can exhibit different colors such as red, blue, green or brown (Figure 2A) [9,10]. The different light-matter interaction at the nano-level for gold is due to the localized surface plasmon resonance (LSPR) phenomena at the nanoscale, which is the collective oscillation of excited electrons in the conduction band of the metal upon light irradiation. LSPR is the origin of the nanogold’s strong optical absorption and scattering at the resonant wavelengths [11]. It is worth mentioning that these nanoparticles have exceptionally high optical absorptivity which typically exceeds 10^5^ times that of common chromophores (extinction coefficients of ~10^9^ M^−1^·cm^−1^), which allows the detection of extremely low concentrations of GNP and thus the development of ultra-sensitive optical-based sensing platforms [12,13]. Encapsulation of “plasmonic” GNP into organic PLGA nanocarriers can thus yield a “Plasmonic composite” that exhibits all the optical features of encapsulated GNP.

The tunability of the GNP’s optical properties can be easily achieved through physiochemical manipulation [7]. Moreover, their optical properties are dependent on the medium refractive index and their aggregation state, which can be used to design highly sensitive detection and sensing platforms (optical response upon the presence of analyte due to aggregation or de-aggregation events) [7]. An old example of employing the brilliant optical properties of GNP for sensing is the pregnancy self-testing strip which is based on immunochromatography (antibody-labeled GNP in a lateral flow assay platform), that has been commonly available in community pharmacies for more than 30 years [14]. In fact, the red line that develops in these tests is produced by GNP which assemble upon binding to the pregnancy hormone [15]. Interestingly, a similar approach has been employed for a wide range of antigens and biological markers; here we notably mention the recent development of the lateral flow immunoassay strips for the rapid detection of IgG antibodies against the COVID-19 virus, which are used in the fight against the current COVID-19 pandemic [16]. This involves the ultra-sensitive detection of nucleic acids using GNP that are functionalized with complementary sequences [17] as well as the tracking of single GNP using hyperspectral-enhanced dark field microscopy (Figure 2B,C) [18,19].

The optical signature of GNP in the near-infra red region (NIR, 700 to 1300 nm, where tissue penetration is maximum due to minimum photon absorption and scattering by water, hemoglobin and other biological components) is a true advantage in biological tracking and sensing applications and allows lower interference due to its biological background [13]. Recently, SoRelle, Leba et al. reported on single GNP tracking with high sensitivity and selectivity in ex vivo tissue sections, which allowed for a detailed evaluation of their biodistributions using hyperspectral microscopy with adaptive detection (hyperspectral dark-field microscopy coupled to computer algorithms) [19]. Moreover, gold nanostructures with NIR optical properties enable other modalities of imaging; this includes photothermal imaging, surface enhanced Raman scattering and photoacoustic imaging [20]. Since the principle of these imaging and diagnostics tools is not the focus of this review, readers are directed to more focused reviews on these topics elsewhere [21,22,23,24,25,26].

## 3. Gold Nanoprobes Enables Electron Microscopy-Based High Spatial Resolution Imaging

In addition to their fascinating optical properties, the elemental and electronic structures of GNP are crucial attributes that make them powerful nanoprobes. GNP are electron-dense and thus act as an efficient electron contrast agent for electron microcopy (EM) imaging. GNP can be easily detected in biological compartments and their detailed shape and precise location can be confirmed at the spatial resolution of electron microscopes (lower than the dimensions of the used GNP themselves). As mentioned in the previous section, the old use of immunogold is an example of utilizing GNP as a contrast agent for EM imaging, which enabled a rich understanding of cellular structures and components. Monoclonal antibodies labeled with GNP have been used to detect RNA polymerase II inside the nucleus of living HeLa cells with localization accuracy in the 10 nm scale [27]. Similar to the early work on labeling monoclonal antibodies, GNP have recently been explored as an electron microscopy contrast agent to label polymeric nanocarriers. The rationale is to encapsulate GNP inside the matrix of polymeric nanocarriers and to allow the later visualization using EM imaging modalities. This approach exhibits various advantages: (1) the encapsulated smaller GNP can be embedded in the interior matrix of the polymeric nanocarriers and not on their surface, and thus it is not expected to alter the surface characteristics of the labeled nanocarriers; (2) only a low number of encapsulated GNP are required to track and visualize labeled polymeric nanocarriers; (3) the availability of rich knowledge and various procedures to prepare GNP with tunable size and shape as well as surface chemistry facilitates an efficient encapsulation into polymeric nanocarriers; (4) the well documented chemical stability of GNP is a clear advantage for the long term studies in tracking/visualization of the vehicle; (5) the ability to tune the size and shape of GNP (spheres, rods, cubes etc.) may allow multiplexed EM detection of polymeric nanocarriers labeled with different gold nanostructures; (6) the biocompatibility of GNP is a clear advantage compared to many other inorganic nanoprobes such as quantum dots or silver nanoparticles.

We have reported on the efficient encapsulation of spherical GNP (15 nm) into poly(lactic-co-glycolic acid) (PLGA) nanocarriers (200 nm) to visualize the latter inside the cytoplasmic compartments of HeLa cells (Figure 3), with spatial details that cannot be revealed by conventional alternative probes and imaging tools such as fluorescence microscopy [28]. Luque-Michel labeled PLGA nanocarriers with small GNP probes that enabled the localization of PLGA nanocarriers inside culture cells using the high backscattered electron capability of the encapsulated gold nanoprobes (Figure 3) [29]. Abstiens et al. [30] proposed two labeling modalities for dual tracking and visualization of PLGA carriers: (1) encapsulation of fluorescent dyes into the core; and (2) assembly of 2.2 nm GNP at the surface to enable detection using fluorescence and electron microscopies. The two modalities complemented each other as the former provided a wider field of view with much lower spatial resolution and the latter provided high spatial resolution at the expense of a lower area available for imaging per analysis (i.e., a narrower field of view). Here, we would like to highlight the fact that encapsulation “into the core” of the polymeric nanocarrier could be preferred compared to their surface assembly “at surface level” to avoid any significant alteration of the surface signature of the carrier itself (probing without alteration).

## 4. Gold Nanoprobes Enables Computed Tomography Imaging

In addition to the capability of visualizing GNP using electron microscopy, the dense electronic nature of gold atoms enables significant X-ray attenuation. Computed tomography (CT) imaging, which relies on X-ray attenuation, is one of the most used imaging platforms in clinical practice due to its availability, short scan time, cost-effectiveness and high spatial resolutions. Since tissues and biological fluids have low attenuation capability, contrasting agents (typically iodine-based media) are required for resolved CT imaging in clinical practice. However, it was found that GNP could provide more efficient X-ray attenuation (5.16 for GNP compared to 1.94 cm^2^/g for iodine, at 100 keV) and they thus hold true promise in CT imaging [31,32,33]. Based on this property, targeted gold nanoprobes have been employed to localize cancer using a standard clinical CT scan [34,35]. Kim et al. showed that labeling stem cells (SC) with gold nanoprobes has enabled the in vivo imaging of SC using CT without notable adverse effects to the labeled cells with a detection limit of as low as 2 × 10^4^ cells/mL in vivo [36]. In vitro studies confirmed the capability of GNP to act as a strong contrasting agent for CT imaging of encapsulating PLGA nanocarriers (Figure 4). However, and to the best of our knowledge, there is no example of using GNP to track polymeric nanocarriers in vivo using a similar approach, which should be a target of.

## 5. Gold Nanoprobes Enable Mass Spectrometry-Based Quantification

Inductively coupled plasma mass spectrometry (ICP-MS) is known to be one of the most powerful analytical tools for the analysis of trace metals. Thus, GNP can be quantified in complex biological samples using ICP-MS with high sensitivity and selectivity and low interference and background levels (LOQ 15 pg/mL) [37]. In fact, it has been shown that ICP-MS is able to monitor the uptake of few GNP into a single cell [38]. Wang et al. applied a droplet-chip-time-resolved ICP-MS single-cell analysis system to quantify the number of GNP uptakes by a single cell and draw conclusions regarding the heterogenicity of the cellular uptake of GNP among cells [38]. In another recent report, single-cell isotope dilution analysis using laser ablation ICP-MS was used to quantify the content of silver nanoparticles inside a single microphage. The reported approach resulted in a limit of detection inside a single cell as low as 0.2 fg Ag per cell [39]. Focused reviews on the applications of ICP-MS for metal-based nanoparticles uptake by cells are available elsewhere [40,41,42].

ICP-MS was used to evaluate the biological distribution of pharmaceutical polymeric carriers loaded with metal-based anticancer therapeutics (e.g., cisplatin) or tagged with metal-based molecular probes. Among the few available examples, PLGA nanoparticle distribution into vital organs in rats after intravenous administration was quantified using ICP-MS upon labeling with palladacycles (pallidum-containing) tags [43]. We have reported on the efficient encapsulation of GNP into PLGA, in which we used ICP-MS to determine the average number of encapsulated GNP per single PLGA nanocarrier which was then used to determine the number of PLGA nanoparticles per cell utilizing the low detection limit of ICP-MS analysis for gold [28]. To the best of our knowledge, using encapsulated metallic nanoprobes to follow up the biodistribution of polymeric hosts in vivo using ICP-MS analysis has not been reported and should be the subject of future research [44].

## 6. Labeling Polymeric Nanocarriers with Gold Nanoprobes as Raman Active Tags

Raman spectroscopy is a powerful analytical tool but suffers from weak sensitivity. GNP are excellent enhancers of Raman scattering for nearby Raman-active tags [45] due to the enhanced electrical field near the surface of the excited GNP [24,46,47,48,49,50]. This phenomenon is called surface enhanced Raman scattering (SERS), which enables the detection of analytes with high sensitivity. Upon laser excitation, the oscillating electrons in the conduction band of the metal nanoparticle generate an electrical field that dramatically improves Raman scattering of nearby molecules [7,12,47,50]. In addition to the electrical field-based enhancement, the chemical enhancement that originates from direct interaction with the surface of the metallic nanostructure contributes to the overall signal enhancement. In fact, GNP and other metallic nanostructures were able to support SERS-based detection to the level of a single molecule [48,51]. It is worth mentioning that SERS signal enhancement is a function of the nanoparticle’s size, shape and the distance of the SERS tags from the nanoparticle surface. Excellent reviews covering the fundamentals of SERS and its emerging contribution in chemical sensing and biological applications are available [47,52,53].

“SERS-tagging” is an emerging labeling technique where a “Raman-active molecule” with a unique scattering fingerprint is attached to GNP to allow the tracking and detection of the tagged nanoparticles [48,52,53,54,55,56,57]. For example, Zavaleta et al. used silica nanoparticles encapsulating GNP that are decorated with a monolayer of Raman active molecules to detect the hybrid nanoparticle in vivo using fiber optic-based Raman endoscopy [52,54]. Despite the exciting SERS phenomena of GNP and other metallic nanostructures (silver and copper), little has been done to encapsulate these SERS tags into pharmaceutical polymeric carriers to allow their tracking and detection using Raman spectroscopy, microscopy or endoscopy [58]. In this direction, Strozyk et al. reported on the efficient encapsulation of GNP that are functionalized with Raman active tags into PLGA microparticles and films to enable SERS-based imaging at high spatial resolution within the encapsulating host. The research team did not evaluate the in vitro and in vivo capability of the prepared PLGA-GNP; however, they concluded that this SERS-active nanocomposite can be used to enable biological long-term monitoring with exceptional stability when compared to conventional fluorescence-based imaging alternatives [59]. Despite the few examples describing the use of polymers as a compartment and substrate to improve SERS reproducibility experimentation, we could not find examples on tracking or imaging of SERS-tagged pharmaceutical polymeric carriers in biological settings using SERS microscopy. We expect that in the near future we will witness more studies in this direction due to the stability of SERS tags and the possibility of designing tags with low interference from biological backgrounds [58].

## 7. Encapsulation Approaches and Surface Functionalization of Gold Nanoparticles

In general, reported approaches to encapsulate GNP into PLGA carriers have been adapted from the available techniques that have been developed to encapsulate both molecular therapeutics and dyes into PLGA matrices. PLGA particles are typically prepared via two methods: (1) the emulsion-evaporation method and (2) the nanoprecipitation method (also called the diffusion-precipitation method).

In the emulsion-evaporation method, PLGA is dissolved into a volatile organic solvent that is immiscible with water, which is then emulsified into aqueous phase containing proper emulsifier to form a stable oil-in-water (*o*/*w*) emulsion. Upon the evaporation of the organic solvent, PLGA will be precipitated into nano- or microparticles depending on the used experimental parameters (Figure 5A). To encapsulate GNP using this method, GNP should exhibit good suspendability in the organic solvents of choice such as chloroform, dichloromethane and ethyl acetate. In addition to the emulsion-evaporation method, the nanoprecipitation method provides a faster and simpler route to prepare PLGA carriers encapsulating GNP. As shown in Figure 5B, the nanoprecipitation method implies the use of an organic solvent that is miscible with water, in which both PLGA and hydrophobic GNP are initially dissolved. Upon addition to the aqueous phase, the organic solvent diffuses rapidly into the aqueous phase, and both PLGA and hydrophobic GNP co-precipitate into nano- or microparticles depending on the experimental settings used.

It is worth mentioning that hydrophilic GNP can also be used with the emulsion-evaporation method but require the preparation of *w*/*o*/*w* emulsion (Figure 6) instead of the simpler *o*/*w* emulsion. In this case, GNP should be suspended in the internal aqueous phase, while PLGA is dissolved in the middle organic phase and upon the evaporation of the organic phase, PLGA nano or microcapsules loaded with GNP can be obtained (Figure 6, left route). Furthermore, to the use of pre-prepared GNP, an alternative route is the use of gold precursors (salt) and reducing agents that can be dissolved in the internal aqueous phase and in situ reduction should take place during the evaporation of the organic phase (i.e., gold reduction during the formation of the PLGA carrier resulting in GNP formation inside the PLGA carriers (Figure 6, right route). Despite the wide use of the emulsion-evaporation method, it suffers from various challenges including a long processing time, the need for a suitable emulsifier, batch-to-batch variability and the use of toxic and flammable organic solvents. From our own observation, we found that the nanoprecipitation method in order to encapsulate PLGA-capped GNP into PLGA nanoparticles is highly convenient, fast and a reproducible procedure with nearly 100% encapsulation efficiency. It is worthy of mentioning that each method (emulsion-evaporation and nanoprecipitation) has its own pros and cons as summarized in Table 1, and the best choice is case-dependent.

In both methods, parameters that affect the resulting GNP-encapsulated PLGA nanocarriers include polymer-related parameters (type, molecular weight), GNP-related parameters (size, surface chemistry), formulation parameters (concentration of polymer, concentration of GNP, type/concentration of stabilizer) and process-related parameters (aqueous to organic volume ration, mixing or emulsification mechanics, evaporation method/rate).

We have evaluated both approaches for the encapsulation of GNP into PLGA carriers and, from the research undertaken, we found that the nanoprecipitation method is a faster and more convenient method when compared to the emulsion-evaporation method. However, various encapsulation approaches may result in very different outcomes. We found that the encapsulation of PLGA-capped GNP into PLGA nanoparticles resulted in higher encapsulation efficiencies when the emulsion-evaporation method was employed compared to nanoprecipitation [28,60]. Luque-Michel et al. compared various emulsion-evaporation methods to encapsulate GNP into PLGA carriers and reported the important effects in regards to type of GNP (hydrophilic, hydrophobic) used, the used method (encapsulation pre-formed versus in situ-formed GNP), type of emulsion (*w*/*o*, *w*/*o*/*w*) and type of emulsifying agent on the encapsulation process [61]. The authors reported that the use of in situ-formed GNP in *w*/*o*/*w* is the only method that can result in an excellent encapsulation with 1:1 GNP to PLGA carrier ratio and no “free GNP” or “empty PLGA carrier” as shown in Figure 7 [61].

The hydrophobicity of PLGA implies the need of proper hydrophobication of the surface of GNP to be encapsulated using the nanoprecipitation or the *o*/*w* emulsion-evaporation methods. The hydrophobication of GNP should render these nanoparticles with excellent colloidal stability in organic solvents that are used for the PLGA nanocarrier preparation, as discussed previously. It is worth mentioning that there are various excellent synthesis routes to prepare hydrophobic small spherical GNP (diameter less than 5 nm), including the widely used Brust methods. However, it is intrinsically challenging to prepare larger nanoparticles with tunable size and complex shapes in organic solvents and, in fact, water is the medium where chemists have developed facile protocols to prepared GNP with various shapes and sizes and with a great degree of tunability. To encapsulate these hydrophilic GNP into a PLGA matrix, post-synthesis surface functionalization/hydrophobication is required, which in many cases could be a challenge [62]. Ligand exchange is the most used approach to displace the hydrophilic ligands (e.g., citrate ions or surfactant molecules) on the surface of GNP with thiolated hydrophobic molecules or polymers to induce a successful phase transfer from water to organic phases [62,63,64]. Various chemistries have been used to manipulate and functionalize the surface of GNP, and these are presented in other review papers [63,64,65,66].

One facile approach that enables the efficient encapsulation of GNP is to modify these nanoparticles with brushes of the same polymer, i.e., PLGA. We described the efficient phase transfer of GNP from water to a dichloromethane or chloroform phase that contains thiolated PLGA upon the addition of methanol (Figure 8). Methanol acts as a cosolvent and decreases the interfacial tension between the aqueous and organic phase to facilitate the assembly of the thiolated PLGA brushes and displace the citrate or other surfactants that were originally capping the GNP in water. We found this method working with outstanding efficiency (nearly 100%) with no nanoparticle aggregation. Notably, the transferred GNP can be collected upon the removal of the organic phase and resuspended in the same solvents that dissolve PLGA, as shown in Figure 8. This method allowed us to prepare GNP-encapsulated PLGA nanoparticles with excellent encapsulation efficiencies, as shown in Figure 8.

Interestingly, we noticed that hydrophobication of the surface of GNP ensures high encapsulation efficacies but may not result in homogeneous distribution inside the PLGA nanocarrier. We compared the encapsulation of GNP with three surface chemistries (polyethylene glycol (PEG), polystyrene (PS) and PLGA) into PLGA nanocarriers [28]. Using the nanoprecipitation method, PEG-capped GNP showed poor encapsulation due to their tendency to “partition or escape” into the aqueous media during preparation as shown in Figure 8. Interestingly, PS-capped GNP exhibited good encapsulation efficiency but poor particle-to-particle homogeneity (many empty PLGA particles and many PS-GNP in one particle that were found to be aggregated and precipitated out of solution) as shown in Figure 8. Interestingly, PS-GNP was found to cluster close to the periphery of the PLGA nanocarrier. Moreover, when the surface of GNP was functionalized with PLGA (the same polymer that makes the carrier matrix), we observed excellent encapsulation efficiencies (about 100%) with excellent distribution of encapsulated GNP inside the PLGA carrier, as shown in Figure 8. These results confirm the importance of the proper surface functionalization of GNP to ensure excellent colloidal stability in organic solvents and hence an efficient encapsulation into the PLGA carrier with homogenous distribution inside the host nanoparticles.

The stability of encapsulated GNP in polymeric nanocarriers can be viewed as a: (1) chemical stability, which is the resistance of the encapsulated metallic nanoprobes to be dissolved; and (2) physical stability, which is the resistance of encapsulated GNP to be released from the polymeric host. GNP have excellent chemical stability and resist dissolution in ambient conditions, including biological compartments [7]. It is worth mentioning that this “chemical inertness” was recently challenged by a recent report on an unexpected intracellular dissolution and recrystallization of GNP in cultured cells [67]. However, the described dissolution processes required several months and it was reported for naked GNP and not for encapsulated counterparts in a hydrophobic matrix such as PLGA [67]. In previous studies, we reported that the encapsulation of GNP into PLGA prevented their etching by cyanide ions, indicating the protective role of the hydrophobic matrix [28]. Regarding the physical stability, and to the best of our knowledge, there are no reports evaluating the tendency of encapsulated GNP or other inorganic nanoparticles to “leach out” from a polymeric host. In one report, Abdel-Mottaleb et al. reported a low release of cadmium selenide quantum dots from 100 nm PLGA nanoparticles after 8 h (4%), and a significant release after one week (30%) into an aqueous buffer system [68]. It will be interesting and valuable to systematically address both the chemical and physical stabilities of encapsulated inorganic nanoparticles in polymeric hosts in upcoming evaluations.

## 8. Doping Polymeric Nanotherapeutics with NIR-Absorbing Gold Nanoparticles: Novel Drug Delivery Systems

Despite the outstanding properties of GNP and their wide range of biomedical and pharmaceutical applications (sensing, imaging and treatment), their use as a “principal” drug delivery system is associated with significant intrinsic limitations. First, with no internal reservoirs like liposomal delivery systems or matrix capabilities mimicking polymeric carriers, drugs are mainly attached to the surface of the GNP, resulting in poor drug loading. Moreover, the nature of drug-GNP conjugation either constitutes physisorption or chemisorption. Physisorption-based conjugation employs weak and reversible physical interactions with GNP or capping agents on their surfaces, resulting in unstable conjugates with highly undesired off-site burst release. Chemisorption-based conjugation employs the formation of strong covalent bonding (gold-thiol, amide, ester) which may retard the extent and rate of drug release at the site of action. Alternatively, polyester polymers form excellent matrices for the delivery of various therapeutics with high loading efficiencies and tunable release. An emerging approach is to combine the outstanding properties of both GNP and polymeric nanocarriers via the fabrication of hybrid composites. The photothermal effect of GNP can be utilized to induce local hyperthermia that can induce cancer ablation [7,69,70,71] and promote the release of loaded drugs from the polymeric host on demand. For example, NIR-absorbing hollow gold nanoshells and an anticancer peptide were encapsulated into PLGA nanocarriers to fight cancer with two strategies: photothermal ablation by GNP and cancer immunotherapy from the encapsulated peptide [72]. The release of the encapsulated peptide was maintained for up to 40 days with excellent photothermal antitumor outcomes using in vivo evaluations. The photothermal effect of encapsulated GNP was shown to contribute towards the killing of cancer cells, as well as to promote the release of the encapsulated therapeutic peptide [73]. In another example, doxorubicin and gold nanorods were both encapsulated into a PLGA matrix to fabricate a thermo-responsive carrier that was able to reduce the size of HeLa spheroids by up to 48% upon laser irradiation [74]. It is worth mentioning that, in addition to the advantage of fighting cancer using two mechanisms, it is advantageous to encapsulate the two “fighters” in the same carrier to ensure identical or similar pharmacokinetics and presence in the site of action. Moreover, nanoparticles may support passive (the enhanced retention and permeation effect) and active targeting, resulting in higher concentration in the tumor. For example, Hao et al. reported on the fabrication of targeted PLGA nanocarriers encapsulating both docetaxel (chemotherapy agent) and gold nanoshells (phototherapy agent). The bioavailability of encapsulated docetaxel into the PLGA carriers increased 1.42 fold in comparison to a solution of free docetaxel [33]. Moreover, laser irradiation at 808 nm resulted in considerable synergistic tumor inhibition efficiency.

## 9. Concluding Remarks and Future Directions

Fabrication of composite nanomaterials has the advantage of bringing two or more materials into one platform that combines the special properties of each. A nanocomposite that is fabricated from FDA-approved polymer and plasmonic nanoparticles is interesting from a scientific and application perspective. These plasmonic polymeric nanoparticles could be easily and accurately quantified, visualized and imaged in complex biological media; hence, allowing for the development of a single platform to fight cancer using loaded chemotherapeutics and the intrinsic photothermal activity of encapsulated gold nanostructures. Various chemical routes are available to prepare these nanocomposites with excellent encapsulation efficiencies. However, there are important gaps that remain to be addressed, such as a thorough characterization of the resulting nanocomposite in terms of proportion of empty polymeric nanocarriers, proportion of free GNP, homogeneity of the encapsulation at the level of the host particle as well as the stability of the nanocomposite itself. Moreover, encapsulation of “renally clearable” plasmonic nanostructures and their complete elimination along with proven degradability of the polymeric host is an attractive direction for research in the near future. Despite the absence of reports investigating the acute and chronic adverse effects associated with labeling biocompatible polymeric nanoparticles with metallic nanoprobes, it is a clear prerequisite for their true clinical and biological applications. Similar to the examples and approaches discussed in this review, the use of other FDA-approved polymers to encapsulate gold and other metallic nanostructures is expected to emerge as stimuli-responsive, thermosensitive, drug-bearing carriers for image-guided targeted delivery and on-site controlled release.

## Figures and Tables

**Figure 1 pharmaceutics-14-00660-f001:**
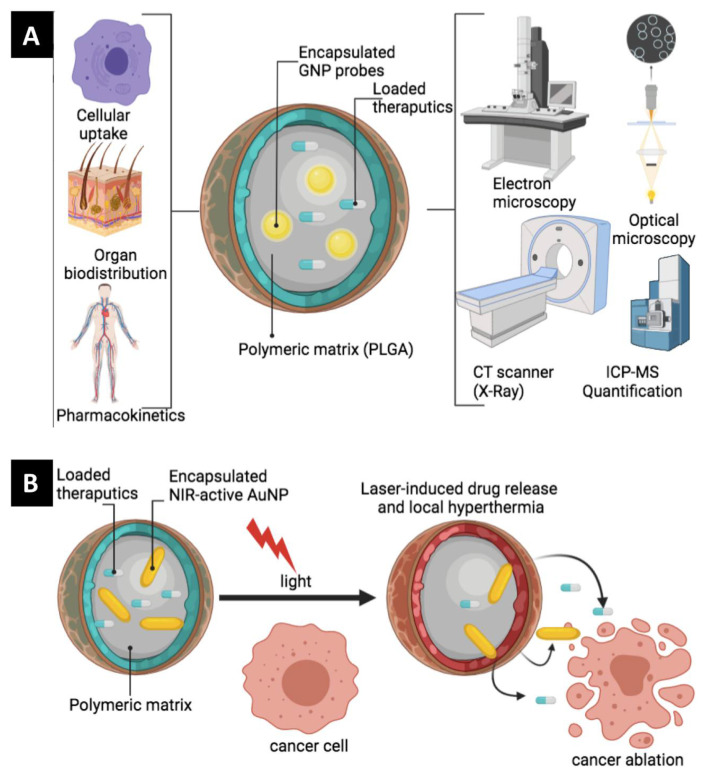
The collective rationale of encapsulating GNP into poly(lactic-co-glycolic acid) (PLGA) nanocarriers: (**A**) Utilizing the optical and electronic properties of GNP to enable the use of various tracking and visualization tools including electron and optical microscopies, ICP-MS and CT-scanning. The labeled PLGA nanocarrier can then be tracked and quantified with accuracy and precision to understand the nano-bio interaction of the PLGA host at the level of cell, tissue and whole organism. (**B**) The photothermal property of encapsulated GNP can be utilized in the development of a light-responsive drug delivery system as well as the synergetic photothermal-chemotherapeutic activity against cancer.

**Figure 2 pharmaceutics-14-00660-f002:**
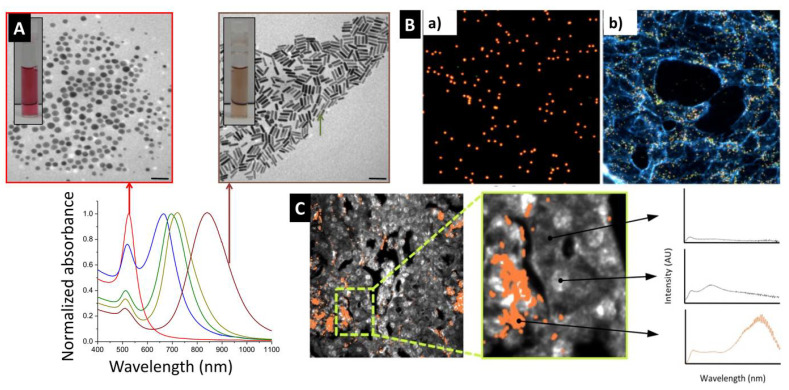
(**A**) Tunable optical properties of GNP. While suspension of spherical GNP (18 nm in diameter) exhibit a red color and plasmonic absorption peak around 520 nm, suspension of gold nanorods exhibit brown color with tunable optical extinction in the visible-infrared region of the spectrum. (**B**) Dark field image of GNP in solution (**a**) and inside cells (**b**). (**C**) Hyperspectral microscopy with adaptive detection analysis showing the location of gold nanorods tracked using their unique optical fingerprint (plasmonic optical extinction in the NIR) in orange and grayscale for tissue background. Images in B and C were reproduced from Refs. [18,19], respectively (Creative Commons Attribution 4.0 International Public License (CC-BY 4.0). (**B**) Reproduced from Zamora-Perez et al. [18] which is licensed under a Creative Commons Attribution CC BY 4.0 Internalional License. (**C**) Reproduced from SoRelle et al. [19] which is licensed under a Creative Commons Attribution CC BY 4.0 Internalional License.

**Figure 3 pharmaceutics-14-00660-f003:**
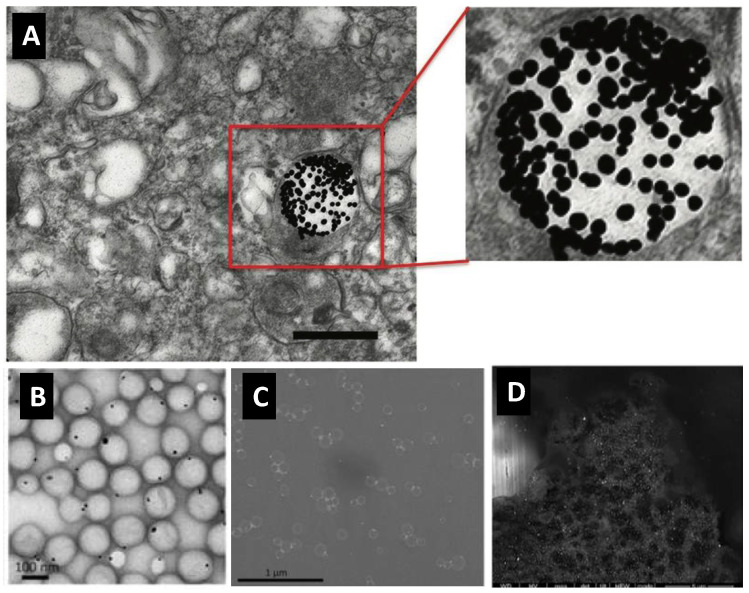
Poly(lactic-co-glycolic acid) (PLGA) nanoparticles labeled with gold nanoprobes to track their nano-bio interaction: (**A**) TEM image showing single PLGA nanocarrier labeled with spherical GNP (15 nm) taken up by HeLa cancer cell, scale bar = 150 nm. (**B**) TEM image and (**C**) SEM backscattered electron image of one-to-one encapsulation of gold nanoprobes into PLGA carriers that allows for the tracking of the labeled PLGA carriers inside J774 cells (**D**) using SEM analysis with backscattered electron detection mode. (**A**) Reproduced with permission from [28], published by John Riley and Sons, 2019. (**B**–**D**) Reproduced with permission from [29] published by Elsevier, 2017.

**Figure 4 pharmaceutics-14-00660-f004:**
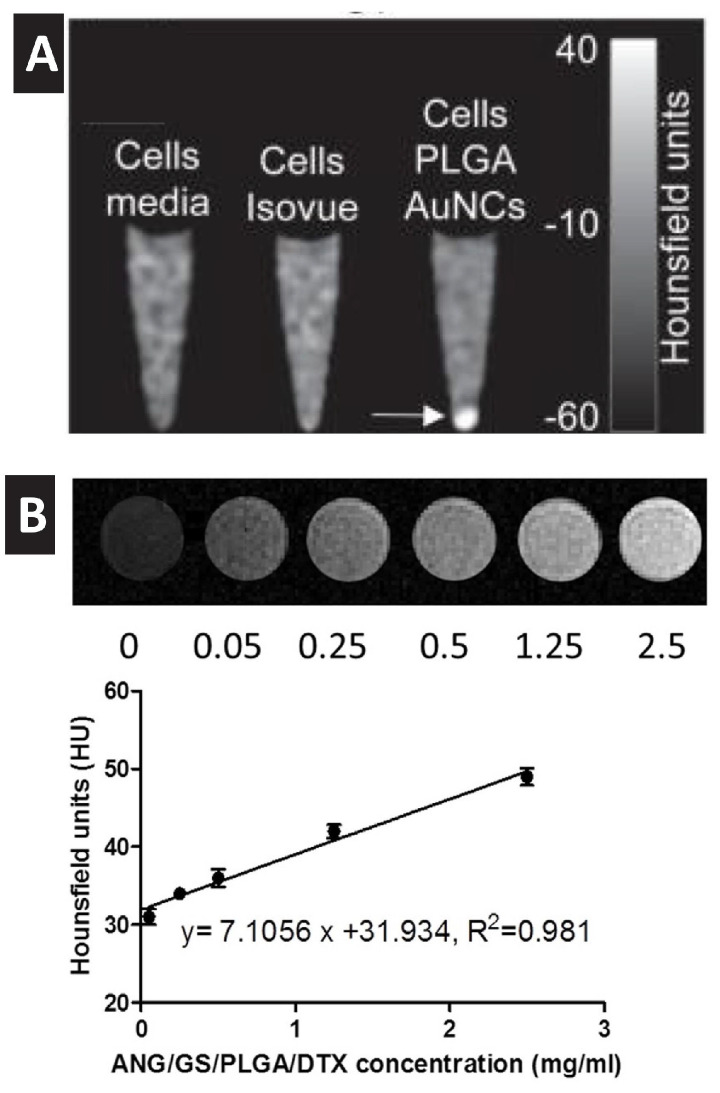
(**A**) In vitro CT image of macrophage cells (J774A) incubated with poly(lactic-co-glycolic acid) (PLGA) nanocarriers encapsulating GNP [arrow]. (**B**) In vitro X-ray CT images [upper panel] and the X-ray attenuation intensity (HU) of GNP loaded PLGA nanoparticles as a function of their concentrations. (**A**) Reproduced with permission from [32], published by Royal Society of Chemistry, 2012. (**B**) Reproduced with permission from [33], published by Elsevier, 2015.

**Figure 5 pharmaceutics-14-00660-f005:**
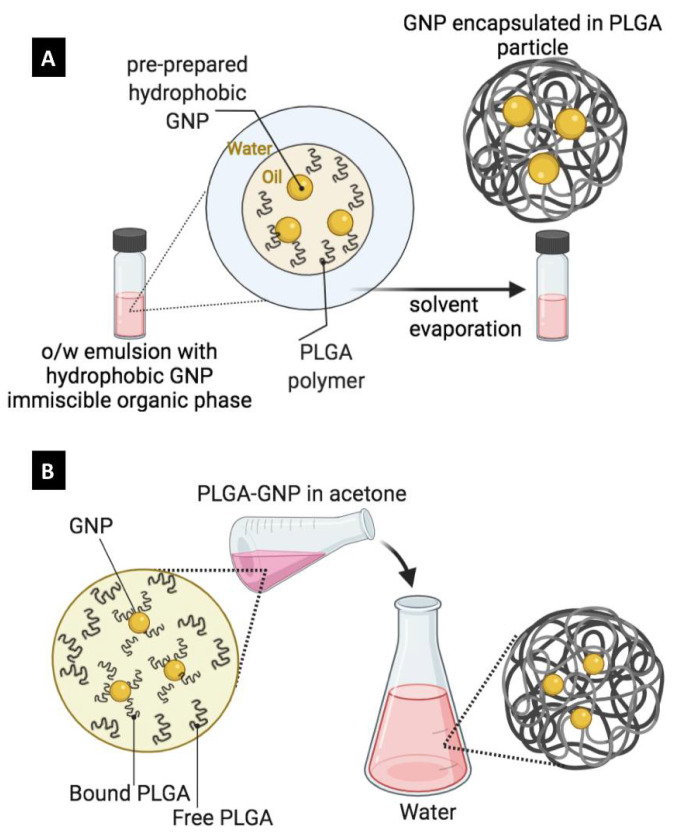
Labeling poly(lactic-co-glycolic acid) (PLGA) particles with GNP. (**A**) *o*/*w* emulsion method in which pre-prepared hydrophobic GNP is suspended in the internal organic phase with PLGA polymers (Emulsion-evaporation method). (**B**) Co-precipitation of pre-prepared hydrophobic GNP and PLGA polymer from the use of a water-miscible organic solvent upon addition to aqueous system (nanoprecipitation method).

**Figure 6 pharmaceutics-14-00660-f006:**
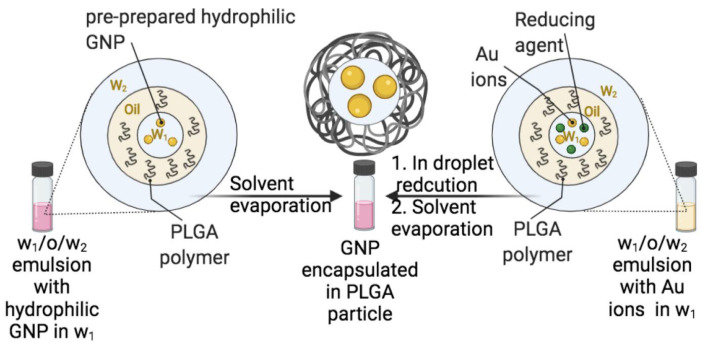
Labeling poly(lactic-co-glycolic acid) (PLGA) nanocarriers with GNP using the *w*/*o*/*w* double emulsion method. Left route: pre-prepared hydrophilic GNP are suspended in the internal aqueous phase. right route: both gold ions (precursor) and the reducing agent are dissolved in the internal aqueous phase. In-droplet reduction and solvent evaporation result in the formation and encapsulation of gold nanoparticles, respectively.

**Figure 7 pharmaceutics-14-00660-f007:**
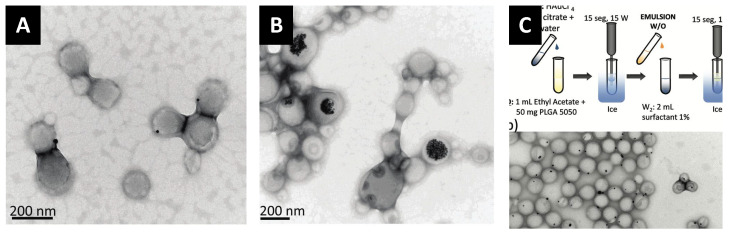
TEM images of poly(lactic-co-glycolic acid) (PLGA) nanoparticles encapsulating GNP prepared using various encapsulation methods. (**A**) Direct encapsulation of citrate-capped GNP in *w*/*o*/*w* emulsion (GNP are in the internal aqueous phase of the double emulsion). (**B**) Direct encapsulation of dodecanethiol-capped GNP in *o*/*w* emulsion (GNP are in the organic phase of the emulsion). (**C**) In situ reduction method in *w*/*o*/*w* emulsion (gold salt and reducing agent are in the internal aqueous phase of the double emulsion). Note that the different encapsulation routes resulted in different encapsulation outcomes; and the 1:1 encapsulation with minimal empty PLGA nanoparticles was obtained only when the in-situ reduction method was employed. Reproduced from Luque-Michel et al. [61] which is licensed under a Creative Commons Attribution CC BY 4.0 Internalional License.

**Figure 8 pharmaceutics-14-00660-f008:**
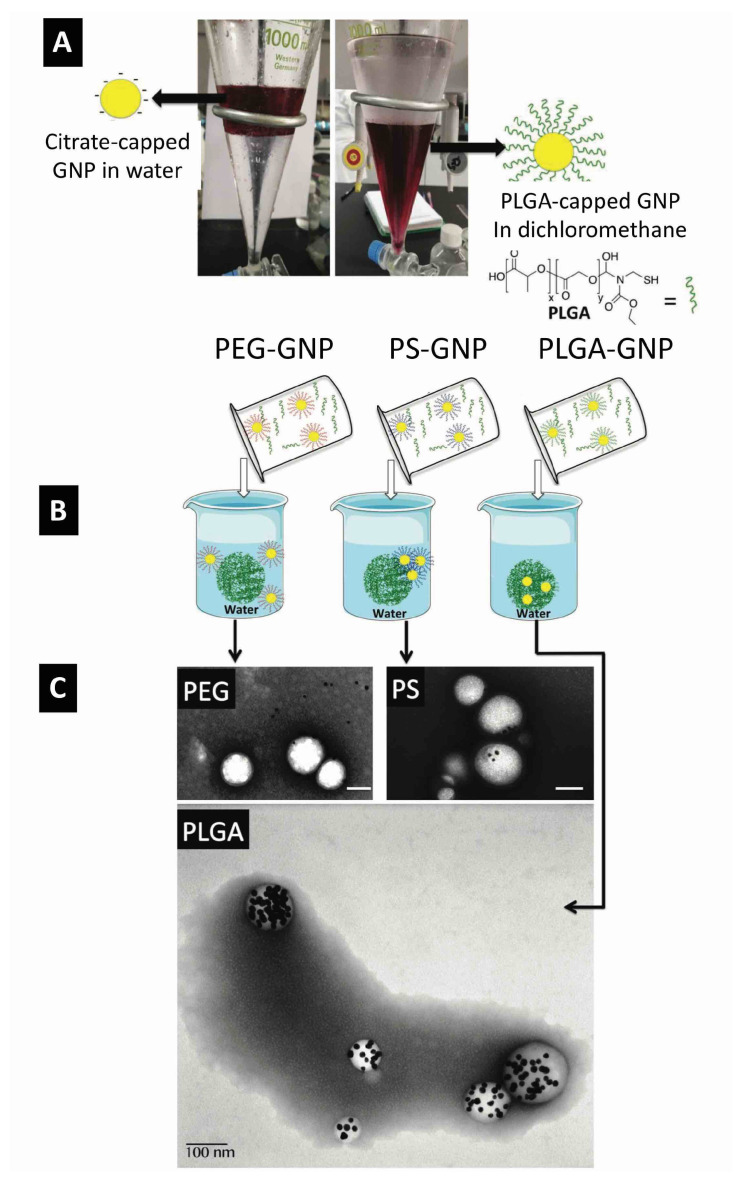
Efficient encapsulation of GNP into poly(lactic-co-glycolic acid) (PLGA) nanoparticles according to the “like dissolves like” principle. (**A**) First, citrate-capped GNP are transferred from water to dichloromethane using PLGA-SH to prepare PLGA-capped GNP that can be re-dispersed in acetone. (**B**) Using the nanoprecipitation method, PEG- or PS- or PLGA-capped GNP are encapsulated into PLGA nanoparticles. (**C**) TEM images conforming the partitioning of PEG-GNP into the aqueous external phase, the poor encapsulation of PS-GNP, and the efficient encapsulation of PLGA-GNP into PLGA nanocarriers. Scale bars are 100 nm in all images. Reproduced from Alkilany et al. [28] which is licensed under a Creative Commons Attribution CC BY 4.0 Internalional License.

**Table 1 pharmaceutics-14-00660-t001:** Comparison between the two major methods to prepare PLGA nano/micro particles.

Emulsion-Evaporation	Nanoprecipitation
Longer and more complex process	Shorter and simpler process
Requires mechanical emulsification to form a stable emulsion	Simple mixing
GNP can be hydrophilic (*w*/*o*/*w* emulsion) or hydrophobic (*o*/*w*)	GNP should be hydrophobic
Hydrophobic GNP should be dissolved in immiscible organic solvents (DCM, chloroform ethyl acetate)	Hydrophobic GNP should be dissolved in miscible organic solvents (acetone, DMSO, DMF, THF)

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
