# Peer review of "PLGA-Gold Nanocomposite: Preparation and Biomedical Applications"

_pharmaceutics, 2022, doi:10.3390/pharmaceutics14030660_

Round 1

Reviewer 1 Report

In this manuscript,the authors summarize the available chemistries and the rationale behind encapsulating GNP into PLGA nanocarriers, and looks into the translation of innovative, clinically applicable nanomedicine. However, some issues should be more discussed in the manuscript.

  1. Page 5,Line 163,what is the meaning of“resistance to itching”?
  2. It would be nice to give some examples about the surface functionalization of gold nanoparticles to help the reader understand more easily.
  3. Please consider adding a table to summarize the biomedical applications, mechanisms or advantages of GNP in the manuscript.

Author Response

Reviewer 1

In this manuscript, the authors summarize the available chemistries and the rationale behind encapsulating GNP into PLGA nanocarriers, and looks into the translation of innovative, clinically applicable nanomedicine. However, some issues should be more discussed in the manuscript.

  1. Page 5,Line 163,what is the meaning of “resistance to itching”?

Response

We thank the reviewer for raising this question. We replaced this sentence: “the well-documented chemical stability and resistance to itching and core dissolution for GNP is a clear advantage for long term tracking/visualization studies” with a shorter version: “the well-documented chemical stability of GNP is a clear advantage for long term tracking/visualization studies”. We believe that the deletion of the part “resistance to itching” makes the sentence clearer and avoid any confusion as suggested by the reviewer’s comment

  1. It would be nice to give some examples about the surface functionalization of gold nanoparticles to help the reader understand more easily.

 Response

To keep the focus of the review paper, we intentionally did not elaborate on the surface functionalization of GNP as a central theme of this review. In fact we directed our readers to excellent review papers that cover this theme in details: “Various chemistries to manipulate and functionalize the surface of GNP are presented in other review papers.[64-67]”

  1. Please consider adding a table to summarize the biomedical applications, mechanisms or advantages of GNP in the manuscript.

Response

We respectfully disagree with this comment. This review focuses on the encapsulation of GNP into PLGA nanocarriers and the biomedical application of the resulting nanocomposite. Thus, the application of GNP alone “in its broad spectrum” is not the focus of this review. We did publish various review papers on the biomedical applications of GNP and all are cited in our review as well. For example, see:

  1. Murphy, C. J.; Gole, A. M.; Stone, J. W.; Sisco, P. N.; Alkilany, A. M.; Goldsmith, E. C.; Baxter, S. C. “Gold Nanoparticles in Biology: Beyond Toxicity to Cellular Imaging” Accounts of Chemical Research 2008, 41, 1721-1730. [Impact Factor* 2010: 852]
  2. Dreaden, E. C.; Alkilany, A. M.; Huang, X.; Murphy, C. J.; El-Sayed, M. A. “The Golden Age: Gold Nanoparticles for Biomedicine” Chemical Society Reviews 2012, 41, 2740-2779 (Invited critical review, most accessed article in ChemSocRev of the month: http://blogs.rsc.org/cs/2012/02/02/top-ten-most-accessed-articles-in-december-2). [Impact Factor*: 44]

Reviewer 2 Report

In the manuscript entitled “PLGA-Gold Nanocomposite: Preparation and Biomedical Applications” the Authors have described review recent advances in the preparation and applications of poly(lactic-co-glycolic acid)-gold nanoparticles (PLGA-GNP) in biomedical aspects. This paper should be revised according to the following comments.

  1. In introduction, part Authors should add pros&cons regarding the application of PLGA-Au NCs system
  2. Regarding the optical properties, the shape and type of AuNPs  e.g. anisotropic, Au-nanorods, nanowires or different Au-based nanostructures should be disused regarding SPR, fluorescence and scattering effects especially in the case of SERS
  3. Please, extend the cytotoxic aspect of the reviewed system: a specially in the context of the cytotoxic mechanism

In my opinion, manuscript required a minor revision before acceptance. 

Author Response

Reviewer 2

In the manuscript entitled “PLGA-Gold Nanocomposite: Preparation and Biomedical Applications” the Authors have described review recent advances in the preparation and applications of poly(lactic-co-glycolic acid)-gold nanoparticles (PLGA-GNP) in biomedical aspects. This paper should be revised according to the following comments.

  1. In introduction, part Authors should add pros&cons regarding the application of PLGA-Au NCs system

Response

We thank the reviewer for this comment. In fact the pros and cons of GNP-PLGA nanocomposite are mentioned clearly in the manuscript. The pros can be found in the introduction while the cons can be found in the conclusion section: “However, there are important gaps still to be addressed such as thorough characterization of the resulting nanocomposite in terms of proportion of empty polymeric nanocarriers, the proportion of free GNP, homogeneity of the encapsulation at the level of host particle as well as the stability of the nanocomposite itself. Moreover, encapsulation of “renally clearable” plasmonic nanostructures and their complete elimination along with proven degradability of the polymeric host is an attractive direction of near-future research.”

  1. Regarding the optical properties, the shape and type of AuNPs  e.g. anisotropic, Au-nanorods, nanowires or different Au-based nanostructures should be disused regarding SPR, fluorescence and scattering effects especially in the case of SERS

Response

We thank the reviewer for this comment. In response, we added this paragraph and three references:

“It is worth to mention that SERS signal enhancement is a function of nanoparticle’s size, shape and the distance of the SERS tag from the nanoparticle surface. Excellent reviews covering the fundamentals of SERS and its emerging contribution in chemical sensing and biological applications are available.[47, 52, 53]

We would like to mention that SERS is not the focus of this review and thus we referred the reader for excellent available recent review on the same topic (Ref 47, 52, 53) in the revised manuscript.

  1. Please, extend the cytotoxic aspect of the reviewed system: a specially in the context of the cytotoxic mechanism

Response

The acute and chronic adverse effects associated with labeling biocompatible polymeric nanoparticles with metallic nanoprobes is not reported in available literature. But the reviewer’s point is important and promoted us to include the following sentence into the conclusion section where we list associated challenges and future directions in the field:

“Despite the absence of reports investigating the acute and chronic adverse effects associated with labeling biocompatible polymeric nanoparticles with metallic nanoprobes, it is a clear prerequisite for their true clinical and biological applications.”

Reviewer 3 Report

This is a very good review. I recommend this work for publication. 

Author Response

Reviewer 3

This is a very good review. I recommend this work for publication. 

Response

We thank the reviewer for his positive feedback.